# Directional-based Wasserstein Distance for Efficient Multi-Agent Diversity

## Abstract

In the domain of cooperative Multi-Agent Reinforcement Learning (MARL), agents typically share the same policy network to accelerate training. However, the use of shared policy network parameters among agents often leads to similar behaviors, restricting effective exploration and resulting in suboptimal cooperative policies. To promote diversity among agents, recent works have focused on differentiating trajectories of different agents given agent identities by maximizing the mutual information objective. However, these methods do not necessarily enhance exploration. To promote efficient multi-agent diversity and more robust exploration in multi-agent systems, we introduce a novel exploration method called Directional Metric-based Diversity (DMD). This method aims to maximize an inner-product-based Wasserstein distance between the trajectory distributions of different agents in a latent trajectory representation space, providing a more efficient and structured Wasserstein distance metric. Since directly calculating the Wasserstein distance is intractable, we introduce a kernel method to compute it with low computational cost. Empirical evaluations across a variety of complex multi-agent scenarios demonstrate the superior performance and enhanced exploration of our method, outperforming current state-of-the-art methods.

## 1 Introduction

Multi-Agent Reinforcement Learning (MARL) has emerged as a promising approach for tackling a variety of multi-agent challenges, including multiplayer video games Vinyals et al. (2019) and autonomous vehicles Cao et al. (2012), drawing increased attention in recent years. MARL enhances collaboration by training multiple agents simultaneously to maximize team rewards. However, challenges persist, such as partial observation restrictions and high scalability demands, which complicate the development of effective cooperative policies for challenging multi-agent tasks. To address these challenges, recent advancements in MARL typically utilize the Centralized Training with Decentralized Execution (CTDE) framework. In this framework, agents take actions based on local observations via decentralized policies that are jointly trained using global information, guaranteeing both robust and stable performance.

With the CTDE framework, each agent develops a decentralized policy, but the training of multiple policy networks can be inefficient. Consequently, parameter sharing is widely adopted, enabling agents to use the same policy network parameters for decision-making. This approach substantially cuts down the number of policy network parameters, thereby decreasing computational costs and accelerating the training process. Furthermore, parameter sharing facilitates the sharing of experiences during centralized training, which enhances robust policy learning and boosts overall efficiency Wang et al. (2020b).

Considering these advantages, a range of MARL algorithms incorporate parameter sharing, including value-decomposition methods Iqbal et al. (2021); Yang et al. (2021); Wang et al. (2020a); Sunehag et al. (2018); Rashid et al. (2018) and policy gradients Ma et al. (2021); Wang et al. (2020d); Ndousse et al. (2021); Zhang et al. (2021). However, the use of shared policy network parameters can result in homogeneous behaviors across agents, which may impede multi-agent diversity and effective exploration Hu et al. (2022). In complex multi-agent environments, extensive exploration and varied policies are vital. For instance, in a football match, agents need to play diverse roles and employ different strategies to collaborate effectively and score goals.

To resolve this problem, prior work proposes to realize identity-aware multi-agent diversity with the maximization of the mutual information between trajectories and identities of agents Jiang & Lu (2021); Li et al. (2021); Rujikorn et al. (2023); Jo et al. (2024). Although these methods successfully learn trajectories that differ from each other, the mutual information metric fails to assess the extent of these differences. Even small differences between trajectories can fulfill the mutual information maximization objective Ozair et al. (2019), which may not effectively promote exploration. To promote exploration, recent advances (Hu et al., 2024; Bettini et al., 2024) have leveraged the maximization of the Wasserstein distance (Villani et al., 2009), a quantity that quantifies the difference between two distributions, to increase policy diversity among agents. In these works, the Wasserstein distance is treated as an intrinsic reward to encourage agents to explore more varied policies. However, these methods fail to account for the similarity in agents' initial policies due to shared policy network parameters, which causes the Wasserstein distance, used to capture policy differences, to converge to zero, rendering the intrinsic rewards ineffective. Moreover, prior works learn diverse policies by simply considering the Wasserstein distance between policies of two agents without accessing a structured or meaningful direction. This may not enable agents to undertake different tasks, leading to chaotic or inefficient cooperation among agents.

To address these limitations and harness the metric-aware advantages of the Wasserstein distance, we introduce a novel exploration method called Directional Metric-based Diversity (DMD), which maximizes an inner-product based Wasserstein distance in a latent contrastive trajectory representation space. To generate meaningful intrinsic rewards using the Wasserstein distance, we propose to learn linearly distinguishable trajectory representations using a novel contrastive loss with learnable identity representations, which serve as linear classifiers, and compute the Wasserstein distance between the trajectory distributions of different agents in the contrastive representation space. To explore the trajectory space more efficiently, we use an inner-product operation to drive the trajectory distributions apart in specific directions, thereby increasing their Wasserstein distance.

The contributions of this work are summarized as follows: (i) Due to the similar initial policies among agents caused by parameter-sharing, which limits the effectiveness of the Wasserstein distance, we introduce a learnable identity representation for each agent and learn a contrastive representation space with a novel contrastive loss to render the Wasserstein distance meaningful. (ii) To calculate the Wasserstein distance efficiently, inspired by the kernel method, we introduce a novel Gaussian kernel method to model the dual function of the Wasserstein distance. (iii) We introduce an inner-product based Wasserstein distance that randomly extends the Wasserstein distance towards every possible direction to efficiently encourage multi-agent diversity. Moreover, such structured exploration enables agents to more likely undertake different tasks, facilitating efficient cooperation. (iv) We further integrate our method with QMIX to implement a practical algorithm. (v) We test our method in challenging tasks from Pac-Men, SMAC Samvelyan et al. (2019), and SMACv2 Ellis et al. (2022) benchmarks. The learning results demonstrate the outperformance of our method compared to state-of-the-art methods.

## 2 BACKGROUNDS

### 2.1 MULTI-AGENT SYSTEM

Consider the model of fully cooperative multi-agent Decentralized Partially Observable Markov Decision Process (Dec-POMDP) Oliehoek & Amato (2015), defined by the tuple $\langle A, S, U, P, R, O, \Omega, \gamma \rangle$, where $A$ represents a group of $|A|$ agents, $S$ denotes the state space, and $U$ represents the action space. Each agent $a$ obtains an observation $o^a \in \Omega$ from the observation function $O(s, a)$ at every time step and chooses an action $u^a \in U$. The actions of all agents combine into a joint action $\boldsymbol{u}$, which prompts the environment to transition to a new state $s'$ according to the transition probability $P(s' \mid s, \boldsymbol{u})$. Concurrently, the agents receive a collective team reward $r = R(s, \boldsymbol{u})$. The discount factor $\gamma \in [0, 1)$ evaluates the importance of future rewards compared to current rewards. The sequence of observation-action pairs $\langle o^a, u^a \rangle$ for each agent $a$ forms its trajectory $\tau^a \in \mathcal{T}$. Each agent $a$ develops its individual policy $\pi^a(u^a \mid \tau^a)$, which together with others, forms a joint policy $\boldsymbol{\pi}$. This joint policy aims to maximize the joint action-value function $Q^{\boldsymbol{\pi}}(s, \boldsymbol{u}) = \mathbb{E}_{s_{0:\infty}, \boldsymbol{u}_{0:\infty}} \left[ \sum_{t=0}^{\infty} \gamma^t r_t \mid s_0 = s, \boldsymbol{u}_0 = \boldsymbol{u}, \boldsymbol{\pi} \right]$.

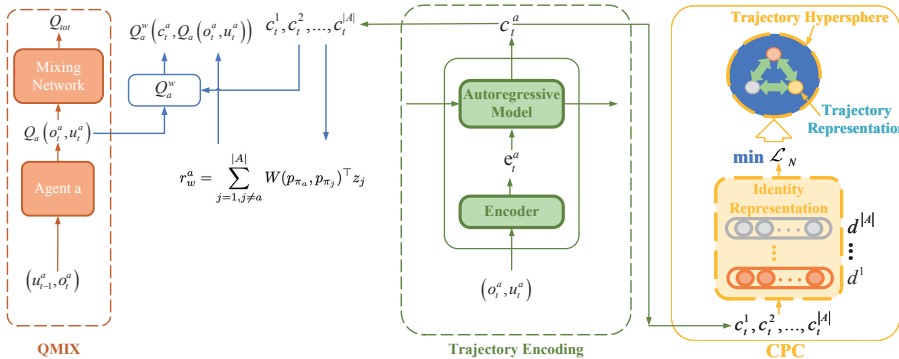

Figure 1: Architecture of DMD. The diagram on the left shows our proposed DMD. We first contrast the learnable identity representations with trajectory representations using Contrastive Predictive Coding (CPC) to learn trajectory representations, enabling the effective use of the Wasserstein distance. We further propose an inner-product based Wasserstein distance, which serves as an intrinsic reward, to encourage efficient multi-agent diversity. The diagram on the right shows our method combined with QMIX, where an additional intrinsic utility network updates agent policies to maximize the intrinsic reward, $r_w^a$.

## 2.2 WASSERSTEIN DISTANCE

The Wasserstein distance addresses an optimal transport problem, quantifying the discrepancy or distance between two probability distributions Villani et al. (2009). For distributions $p$ and $q$ respectively defined over domains $\mathcal{X} \subseteq \mathbb{R}^m$ and $\mathcal{Y} \subseteq \mathbb{R}^n$, we can give the equation of the Wasserstein distance as follows:

$$\mathcal{W}_c(p, q) = \inf_{\gamma \in \Gamma(p,q)} \int_{\mathcal{X} \times \mathcal{Y}} c(x, y) \mathrm{d}\gamma(x, y) \tag{1}$$

where $c(x,y) \colon \mathcal{X} \times \mathcal{Y} \to \mathbb{R}$ is a cost function, $\Gamma(p, q)$ represents the set of all potential couplings of the distributions $p$ and $q$ defined in the product space $\mathcal{X} \times \mathcal{Y}$. The probability distributions $p$ and $q$ serve as the marginal distributions of the coupling $\gamma(x, y)$ in the space $\mathcal{X}$ and $\mathcal{Y}$, respectively.

In practice, a smoothed Wasserstein distance $\tilde{W}_c(p, q)$ is adopted in our method to avoid converging to sub-optimum. Computing $\tilde{W}_c(p, q)$ directly is usually intractable, so we utilize a tractable smoothed Fenchel-Rockafellar duality Villani et al. (2009),

$$\tilde{W}_c(p, q) = \sup_{\mu, \nu} \mathbb{E}_{x \sim p(x), y \sim q(y)} \left[ \mu(x) - \nu(y) - \beta \exp \left( \frac{\mu(x) - \nu(y) - c(x, y)}{\beta} \right) \right] \tag{2}$$

where $\mu : \mathcal{X} \to \mathbb{R}$ and $\nu : \mathcal{Y} \to \mathbb{R}$ serve as dual functions over continuous domains, and $\beta$ represents a smoothing parameter. This dual formulation of the Wasserstein distance enables the parametrization of the dual functions, effectively reducing the computational complexity involved in solving the optimal transport problem.

## 3 DIRECTIONAL METRIC-BASED DIVERSITY

In this section, we detail our proposed Directional Metric-based Diversity (DMD). First, we present how to learn meaningful representations to generate effective feedback for the Wasserstein distance. Then, we show how to maximize the Wasserstein distance between different trajectory distributions in the contrastive representation space.

### 3.1 IDENTITY-AWARE TRAJECTORY REPRESENTATIONS

Agents that share the same policy network parameters start with similar initial policies. As a result, the Wasserstein distance between the policy distributions of any two agents tends to converge

to zero, i.e., $W(X, Y) \to 0$, where $X$ and $Y$ represent the policy distributions of two agents, respectively. To address this problem, we use a contrastive loss to learn linearly distinguishable trajectory representations. This method ensures that trajectories generated by the policies of different agents are mapped to diverse distributions in a latent representation space. Within this space, the Wasserstein distance serves as an effective mechanism for encouraging the learning of diverse policies.

First, we present the structure of the trajectory encoder. Each observation-action pair $x_t^a = (o_t^a, u_t^a)$ is encoded into a latent embedding space $e_t^a = g_{\theta_e}(x_t^a)$ using a non-linear encoder $g_{\theta_e}$. Subsequently, an autoregressive model $g_{\theta_g}$ combines these latent embeddings to derive the trajectory representation $c_t^a = g_{\theta_g}(e_{\leq t}^a)$ at timestep $t$. We refer to the combination of these models as $g_\theta = \{g_{\theta_e}, g_{\theta_g}\}$, representing the entire trajectory encoding mechanism. For simplicity, standard architectures like MLPs are used for $g_{\theta_e}$ and GRUs for $g_{\theta_g}$.

We next use a contrastive loss based on Contrastive Predictive Coding (CPC) Oord et al. (2018) to train the trajectory encoder $g_\theta$ in order to learn linearly distinguishable trajectory representations. Due to the similar trajectory samples generated by homogenous initial policies, employing the vanilla CPC to directly contrast such trajectory samples with each other may not learn distinguishable trajectory representations. To solve this problem, we instead contrast trajectory samples with randomly initialized learnable identity representations. The identity representation $d^a \in \mathbb{R}^H$ for each agent serves as a linear classifier to distinguish trajectory representations of different agents. With the identity representations, we adapt CPC from representational learning with an unsupervised method to that with a supervised method.

Consider a set $\mathcal{C} = \left\{ c_t^{a'} \right\}_{a'=1}^{|A|}$ that includes all agents' trajectory representations at time step $t$, and the identity representation of agent $a$, $d^a$. The objective of CPC is to ensure that $d^a$ stays close to its corresponding trajectory representation while maintaining separation from the other trajectory representations in $\mathcal{C} \setminus \{c_t^a\}$, which can be written as follows:

$$\mathcal{L}_N = - \mathop{\mathbb{E}}_{(d^a, \mathcal{C}) \sim \mathcal{D}} \left[ \log \frac{f\left(c_t^a, d^a\right)}{\sum_{c_t^{a'} \in \mathcal{C}} f\left(c_t^{a'}, d^a\right)} \right] \tag{3}$$

where $f(c_t, d) = \exp\left(c_t^T d\right) \in \mathbb{R}$. The dot product $c_t^T d$ quantifies the similarity between the trajectory representation $c_t^a$ and the identity representation $d^a$. By minimizing the contrastive loss, our method trains both the trajectory encoder $g_\theta$ as well as the identity representation $d^a$ and enforces trajectory representations to distribute around their corresponding identity representations that are uniformly distributed in the trajectory representation hypersphere. As a result, with linear identity representations, the trajectory representations of different agents are linearly classified for the minimal contrastive loss.

Some recent methods also adopt the contrastive learning method for credit assignment Liu et al. (2023), learning distinguishable trajectory representations Li et al. (2024), and efficient exploration Li & Zhu. However, different from these works, we employ contrastive learning specifically to render the Wasserstein distance meaningful. The minimization of the contrastive loss makes the trajectory representations distinguishable, enabling the Wasserstein distance that measures the differences of trajectory distributions to be effective.

## 3.2 INNER-PRODUCT BASED WASSERSTEIN DISTANCE

We next encourage visitations of diverse trajectories by maximizing the Wasserstein distance between different agents' trajectory distributions in a latent representation space. We first present how to efficiently calculate the Wasserstein distance between two trajectory distributions. Then we provide a novel inner-product based Wasserstein distance to encourage multi-agent diversity.

Let $p_{\pi_1}$ and $p_{\pi_2}$ represent the distributions of trajectory representations for agent 1 and agent 2, respectively. The definition of the Wasserstein distance between $p_{\pi_1}$ and $p_{\pi_2}$ is as follows:

$$\tilde{W}_c(p_{\pi_1}, p_{\pi_2}) = \sup_{\mu, \nu} \mathbb{E}_{c_t^1 \sim p_{\pi_1}, c_t^2 \sim p_{\pi_2}} \left[ \mu(c_t^1) - \nu(c_t^2) - \beta \exp\left( \frac{\mu(c_t^1) - \nu(c_t^2) - c(c_t^1, c_t^2)}{\beta} \right) \right] \tag{4}$$

where the cost function $c(c_t^1, c_t^2)$ is defined by the Euclidean distance between the points $c_t^1$ and $c_t^2$, specifically $c(c_t^1, c_t^2) = \|c_t^1 - c_t^2\|$. The calculation of the Wasserstein distance refers to the optimization over dual functions towards maximizing Equation 4. In multi-agent settings, as each agent needs

to calculate the Wasserstein distance between itself and other agents, simply parameterizing dual functions with neural networks like previous works He et al. (2022); Park et al. (2024) may incur high computational costs. We instead employ the kernel method, commonly utilized in machine learning Hearst et al. (1998). In particular, we represent the dual functions as linear combinations of Gaussian kernel functions, approximated using random feature maps Rahimi & Recht (2007). For example, the dual function $\mu$ is given by $\mu(\mathbf{x}) = (\lambda^\mu)^\top \phi(\mathbf{x})$. Here, for $\mathbf{x} \in \mathbb{R}^d$, $\phi(\mathbf{x}) = \frac{1}{\sqrt{m}} \cos(\mathbf{Gx} + \mathbf{b})$ denotes an $m$-dimensional random feature map, with $\mathbf{G} \in \mathbb{R}^{m \times d}$ being a Gaussian matrix consisting of entries drawn from a normal distribution $\mathcal{N}(0, 1)$ and $\mathbf{b} \in \mathbb{R}^m$ containing entries drawn from a uniform distribution $U(0, 2\pi)$. This implies that during the optimization of the dual function $\mu$, it is only necessary to update the dual vector $\lambda^\mu \in \mathbb{R}^m$, thus significantly decreasing the computational demand compared to using computationally intensive neural networks for parameterizing dual functions.

To derive the optimal dual functions, we utilize stochastic gradient descent (SGD) on the Wasserstein distance objective outlined in Equation 4. The dual functions $\mu$ and $\nu$ are represented by the kernels $\kappa$ and $\ell$, respectively. With trajectory representation samples $\{c_t^1, c_t^2\}$ from distributions $(p_{\pi_1}, p_{\pi_2})$, we apply the chain rule to Equation 4, deriving the gradients with respect to $\lambda^\mu$ and $\lambda^\nu$ as follows:

$$
\nabla_{(\lambda^\mu, \lambda^\nu)} \tilde{W}_c(p_{\pi_1}, p_{\pi_2}) = \mathbb{E}_{c_t^1 \sim p_{\pi_1}, c_t^2 \sim p_{\pi_2}} \left[ (1 - x) \begin{pmatrix} \phi_\kappa(c_t^1) \\ -\phi_\ell(c_t^2) \end{pmatrix} \right],
$$
$$
where \quad x = \exp \left( \frac{(\lambda^\mu)^\top \phi_\kappa(c_t^1) - (\lambda^\nu)^\top \phi_\ell(c_t^2) - C(c_t^1, c_t^2)}{\beta} \right) \tag{5}
$$

We estimate the expectation by computing the average of function values across a batch of trajectory representation samples that are sampled from the replay buffer, which stores the agent's experiences during training.

Since we have achieved the value of the Wasserstein distance, we now present our inner-product based Wasserstein distance given by

$$
r_w^a = \sum_{j=1, j \neq a}^{|A|} W(p_{\pi_a}, p_{\pi_j})^\top z_j. \tag{6}
$$

To integrate our method with existing MARL methods, we treat $r_w^a$ as an intrinsic reward for agent $a$. This intrinsic reward intuitively aligns the directions of the Wasserstein distance between the trajectory distributions of different agents with the latent variable $z_j$. Here, the latent variable $z_j \in \mathbb{R}^M$, assigned to agent $j$, is randomly sampled from a fixed uniform distribution $p(z)$. Maximizing the intrinsic reward $r_w^a$ using MARL methods enables the trajectory distribution of the current agent $a$ to increase the Wasserstein distance from those of other agents along directions aligned with random latent variables $z$, thereby resulting in the visitation of diverse trajectories with significant variations. Such directional movements also lead to the efficient exploration of the trajectory space.

Compared to simply maximizing the sum of the Wasserstein distances without the latent variable $z$, our method encourages multi-agent diversity in a more structured way, which empirically achieves better performance, as verified by the ablation study results shown in Figure 4a. This is because the latent variable $z_j$ for each agent $j$ provides meaningful signals about how the current agent should be different from other agents, i.e., agents are more likely to undertake different tasks. Without a structured or meaningful direction, agents might learn to be different in arbitrary or unproductive ways, which may not benefit the cooperation of multiple agents.

### 3.3 PRACTICAL LEARNING ALGORITHM

We next demonstrate how to incorporate our method into QMIX Rashid et al. (2018), a value-decomposition based MARL algorithm. QMIX optimizes individual policies for agents by learning the optimal joint action-value function $Q^{\boldsymbol{\pi}}$, which is estimated by $Q_{tot}$, a mixing network that monotonically combines the utilities of all agents (from which the policies are derived). To incorporate our method into QMIX, we introduce an additional intrinsic utility network $Q_a^w$ (The reason for why

we use the intrinsic utility network is discussed in Appendix E). This network takes the individual agent utility $Q_a(o_t^a, u_t^a)$ as well as the trajectory representation $c_t^a$ as inputs. We learn the optimal $Q_a^w$ to maximize the Wasserstein distance-based intrinsic rewards provided by our method by minimizing the TD loss:

$$\mathcal{L}_{TD}^w = \mathbb{E}_{(o_t^a, u_t^a, o_{t+1}^a) \sim \mathcal{D}} \left[ \left( Q_a^w \left( c_t^a, Q_a(o_t^a, u_t^a) \right) - y \right)^2 \right],$$
$$where \quad y = r_w^a + \gamma \bar{Q}_a^w \left( c_{t+1}^a, \bar{Q}_a \left( o_{t+1}^a, u_{t+1}^a \right) \right) \tag{7}$$

where $\bar{Q}_a^w$ and $\bar{Q}_a$ represent target networks used to stabilize training, while $\mathcal{D}$ denotes the replay buffer. The TD loss, $\mathcal{L}_{TD}^w$, functions as a regularizer, adding an auxiliary gradient to the agent utility network $Q_a$ to facilitate the learning of diverse trajectories. Consequently, we can formulate the total loss function as follows:

$$\mathcal{L}_{total} = \mathcal{L}_{TD} + \alpha \mathcal{L}_{TD}^w \tag{8}$$

where $\mathcal{L}_{TD}$ represents the TD loss used in QMIX to learn the optimal $Q_{tot}$, and $\alpha$ is a coefficient adjusting the weight of $\mathcal{L}_{TD}^w$. Our method gradually converges to QMIX as the cofficient $\alpha \to 0$. By minimizing $\mathcal{L}_{total}$, the policies of all agents are jointly trained end-to-end in order to maximize both team rewards and the Wasserstein distance between the trajectory representation distributions of different agents. The pseudocode of our method is provided in Appendix G. We also implement our method on top of MAPPO, a policy-based method. We refer the reader to Appendix H for more details.

A potential concern is that the additional exploration from the regularizer $\mathcal{L}_{TD}^w$ could affect the convergence of the integrated MARL method. We provide a theoretical analysis of our method's convergence guarantee in Appendix F.

## 4 EXPERIMENTS

In experiments, we evaluate our method using challenging multi-agent tasks from Pac-Men, SMAC, and SMACv2 to highlight its effectiveness. We conduct a comparative analysis against state-of-the-art MARL methods, including value-decomposition methods (QMIX Rashid et al. (2018) and QTRAN Son et al. (2019)), mutual information-based exploration methods (MAVEN Mahajan et al. (2019), EOI Jiang & Lu (2021), SCDS Li et al. (2021), PMIC Li et al. (2022), LIPO Rujikorn et al. (2023), and FoX Jo et al. (2024)), and Wasserstein distance-based diversity methods (MAPD (Hu et al., 2024) and DiCo (Bettini et al., 2024)). For generality, we present results with both the mean and standard deviation of performance, tested over five random seeds. Training details and hyperparameters are reported in Appendix N. The source code of our method can be found in the supplementary material.

### 4.1 PAC-MEN

We first examine our method in Pac-Men to evaluate its effectiveness in promoting diversity among multiple agents. In Pac-Men, as shown in Figure 2a, four agents start in the central room of a maze. Each agent can only see a 4×4 grid around them. The edge rooms of the maze contain randomly placed dots. The agents' goal is to collect as many of these dots as possible. We modify the lengths of paths leading to the edge rooms to evaluate the exploration capacities of agents. Notably, when agents stay in the same edge room, it can lead to inefficient competition. Ideally, they should perform varied behaviors and move to different rooms.

The results depicted in Figure 2b highlight our method's superior performance compared to baseline methods. By maximizing the Wasserstein distance among trajectory distributions in a structured way, agents trained by our method distribute themselves in the four edge rooms, as illustrated in Figure 2d. This phenomenon demonstrates that our method promotes diverse policies, leading to efficient cooperation. MAPD struggles to learn diverse policies, as illustrated in Figure 2c, where some agents learn identical policies and go to the same edge room, leading to suboptimal performance. Additionally, Figure 2e reveals that MAPD's intrinsic rewards, based on the Wasserstein distance in the raw trajectory space, are ineffective in encouraging the exploration of diverse policies. In contrast, our method leverages a contrastive loss with identity representations to learn a contrastive trajectory representation space, resulting in a more effective utilization of the Wasserstein distance as

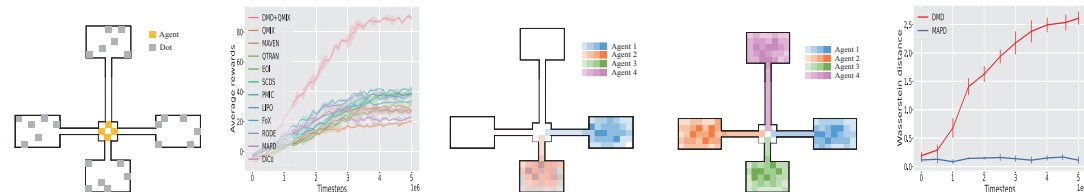

(a) Illustration of Pac-Men  (b) Average rewards  (c) Heatmap of MAPD  (d) Heatmap of DMD  (e) Wasserstein distance

Figure 2: Comparisons of performance between our proposed DMD and baseline methods in Pac-Men. We present both the mean and standard deviation across five random seeds.

intrinsic rewards. Moreover, the inner-product operation leverages the latent variables to efficiently drive agents into different edge rooms.

Some baselines that use mutual information, like EOI and SCDS, and incorporate variational intrinsic rewards, result in very similar performance. These methods often fail to locate the edge room with the longest path since the variational intrinsic reward fails to induce efficient exploration, resulting in suboptimal performance. The quick convergence of the variational intrinsic reward, owing to its metric-agnostic property, does not sufficiently incentivize exploration. In contrast, our method's Wasserstein distance-based, metric-aware intrinsic reward consistently delivers effective reward signals that promote efficient exploration.

## 4.2 SMAC

We subsequently examine our method in the StarCraft Multi-Agent Challenge (SMAC) Samvelyan et al. (2019), which is a widely used benchmark for testing cooperative MARL algorithms. SMAC consists of a variety of combat scenarios with different levels of difficulty. Our evaluation covers six scenarios within SMAC, namely: 3s5z (easy), 2c_vs_64zg (hard), 7sz (hard), 6h_vs_8z (super hard), corridor (super hard), and 3s5z_vs_3s6z (super hard). We utilize version SC2.4.10 of SMAC for our experiments. Note that performance comparisons between different versions of SMAC are not applicable.

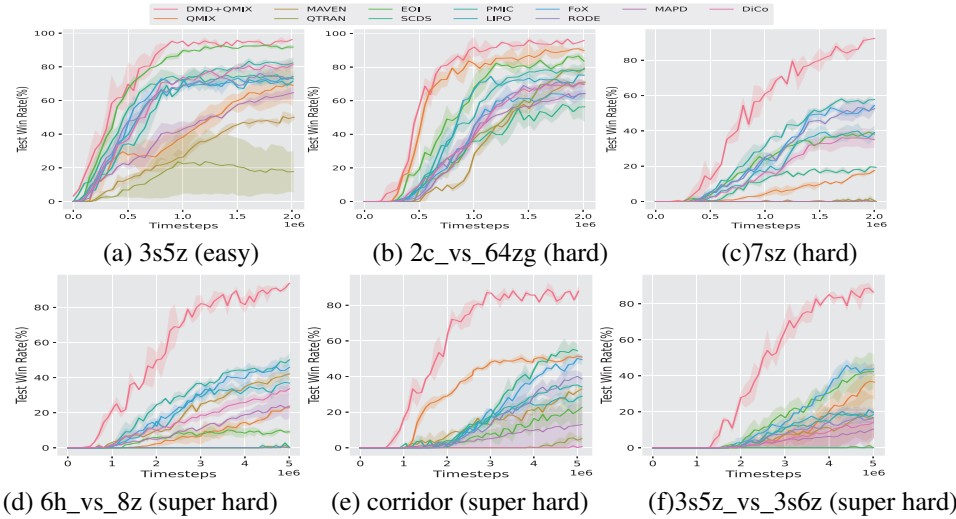

(a) 3s5z (easy)  (b) 2c_vs_64zg (hard)  (c)7sz (hard)

(d) 6h_vs_8z (super hard)  (e) corridor (super hard)  (f)3s5z_vs_3s6z (super hard)

Figure 3: Comparisons of performance between our proposed DMD and baseline methods in the scenarios of SMAC.

As illustrated in Figure 3, our method not only performs well in both easy and hard scenarios but also significantly surpasses all baseline methods in super hard scenarios. QMIX encounters

difficulties in developing complex cooperative policies required in super hard scenarios. However, our method enhances QMIX's performance by efficiently promoting diversity among multiple agents. Furthermore, our method achieves a significant performance improvement over MAPD, demonstrating the effectiveness of maximizing the inner-product based Wasserstein distance within the contrastive trajectory representation space. Notably, MAPD performs even worse than mutual information-based methods in extremely challenging scenarios. This outcome can be attributed to the limitations of the Wasserstein distance metric used in MAPD, which may not function effectively under the policy network parameter-sharing setting.

Compared to the baselines based on mutual information, our method shows superior results by maximizing the metric-aware Wasserstein distance, which fosters more diverse trajectories and leads to more efficient exploration. We provide some visualization examples of diverse policies learned by our method in Appendix 8. The baselines based on mutual information often fail to enable agents to learn trajectories with significant variations. EOI does not achieve satisfactory results because the trajectory classifier, employed to distinguish trajectories of different agents, tends to overfit to agent identity information, thus limiting exploration. MAVEN is less efficient to search for cooperative policies since agents often learn static joint behaviors with small differences.

**Homogeneous behaviors** In 3s5z, agents require to master the trick of 'focus fire', where agents behave similarly to target and fire at the same enemy. Our method successfully learns 'focus fire' and achieves satisfactory performance in 3s5z, demonstrating that our method would not impede homogeneous behaviors if they can lead to more environmental rewards. We refer the reader to Appendix I.2 for more evaluations of our method in homogeneous scenarios.

To examine the impact of stochasticity from environments on the performance of our method, we also test DMD in a stochastic benchmark SMACv2. The results are provided in Appendix I.2. More evaluations of our method in Google Research Football (GRF) are also provided in Appendix I.2.

### 4.3 ABLATION STUDY

We take several ablation studies to examine the key components within our method. To test the autoregressive model in learning trajectory representations, we remove this model, relying solely on the non-linear encoder $g_{\theta_e}$ without considering trajectory context. To examine our CPC, we introduce five variations: (i) using an encoder that is randomly initialized with fixed parameters for trajectory encoding, (ii) directly predicting agent identities of different trajectories by minimizing a cross-entropy loss rather than the contrastive loss to learn trajectory representations, (iii) using fixed agent identities such as randomly initialized one-hot vectors instead of learnable identity representations in the contrastive loss to learn trajectory representations, and (iv) like the vanilla CPC, we directly contrast trajectory samples with each other rather than the identity representations to investigate whether the vanilla CPC can learn distinguishable trajectory representations. To evaluate the effectiveness of our Wasserstein distance objective, we ablate the inner-product based Wasserstein distance $r_w^a$, and instead use the sum of the Wasserstein distances between the trajectory representation distribution of the current agent and those of other agents without considering the latent variables. We also use the Wasserstein distance between the current agent and a randomly chosen agent.

We evaluate these variants using three scenarios from SMAC, and the results are presented in Figure 4a. We observe that ablating any components from our method significantly hurts performance. Notably, as task complexity increases, the absence of the autoregressive model leads to noticeable declines in performance, indicating the importance of learning trajectory representations for robust performance in challenging environments. Employing a fixed encoder for encoding trajectory representations results in suboptimal performance, underscoring the necessity of using CPC to generate distinguishable trajectory representations. The variants that focus on the identity prediction or use fixed agent identities in the contrastive loss prove to be less effective. Moreover, using the vanilla CPC results in significant performance drop, demonstrating the effectiveness of identity representations used in CPC to distinguish trajectory representations of different agents.

The variant using the sum of Wasserstein distances as an intrinsic reward does not yield satisfactory results and even performs worse than the variant based on the random agent Wasserstein distance. Figure 4b illustrates that the sum of Wasserstein distances does not effectively promote multi-agent diversity. In contrast, our inner-product based Wasserstein distance can continuously power efficient

multi-agent diversity since the latent variables can efficiently promote the structured exploration of trajectory space. We further present the UMAP visualizations of trajectory representations in Figure 5. The UMAP of DMD demonstrates the alignment of trajectory representations with the latent variables. However, without the latent variables in the objective, the trajectory representations become chaotic and misaligned. Despite the performance degradation induced by different implementations of the Wasserstein distance, these variants still significantly outperform QMIX, highlighting the robustness of our representation learning method.

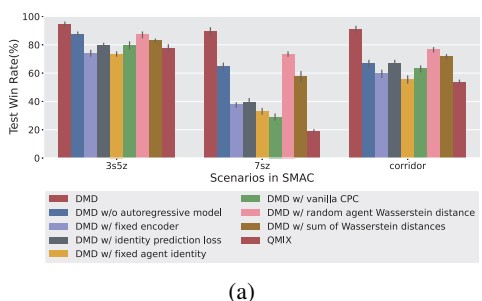
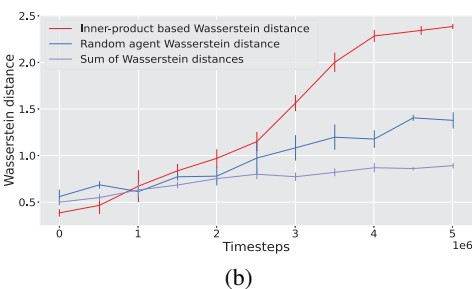

(a)    (b)

Figure 4: (a) Performance comparisons between our proposed DMD and different variants in the scenarios of SMAC. (b) Various types of Wasserstein distances.

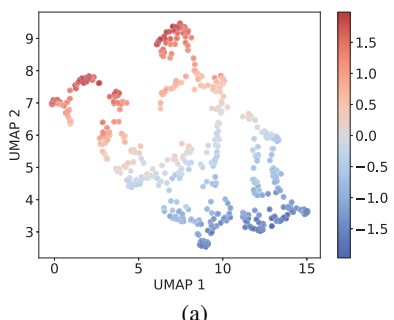
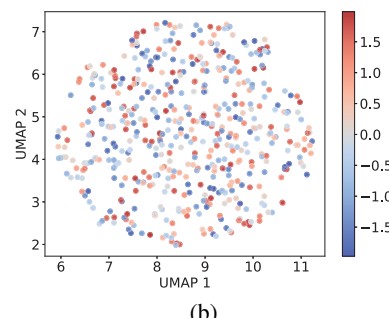

(a)    (b)

Figure 5: UMAP visualizations of trajectory representations of (a) DMD with latent variables and (b) DMD without latent variables

## 5  RELATED WORKS

We discuss related work on multi-agent diversity and Wasserstein distance. Due to page limits, we have placed this discussion in Appendix B.

## 6  CONCLUSION

In this paper, we introduce a novel exploration method, namely DMD. Different from prior Wasserstein distance-based methods, our method proposes an inner-product based Wasserstein distance between different agents' trajectory distributions in a latent trajectory representation space, leading to the visitations of more diverse trajectories and efficient structured exploration. We learn meaningful trajectory representations for measuring the Wasserstein distance using a novel contrastive loss with learnable identity representations. We incorporate our method into QMIX by developing an intrinsic utility network that focuses on maximizing intrinsic rewards based on the inner-product based Wasserstein distance. We evaluate DMD on a variety of challenging multi-agent tasks and the results demonstrate our method's superior performance and efficient exploration.

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

# A    APPENDIX

# B    RELATED WORKS

**Multi-agent diversity** To promote multi-agent diversity, numerous methods have been proposed that incorporate various intrinsic motivations or regularizers. RODE Wang et al. (2020c) develops diverse policies by allocating different actions to predefined roles, but this method might be less effective in environments with continuous actions and large action spaces. MAVEN Mahajan et al. (2019) employs a value-based strategy that associates latent variables, managed by a hierarchical policy, with the joint behaviors of agents by the mutual information objective. EOI Jiang & Lu (2021) adopts a supervised learning technique to enhance agent individuality, using a probabilistic classifier to model the probability distributions of agents based on their observations. SCDS Li et al. (2021) aims to promote multi-agent diversity by maximizing the objective of the mutual information between agent identities and their trajectories. PMIC Li et al. (2022) promotes visits to different cooperative behaviors by maximizing mutual information related to superior cooperative behaviors while minimizing it for less effective ones. LIPO Rujikorn et al. (2023) proposes using policy compatibility to learn varied policies and promotes diverse behaviors of agents via the mutual information objective. FoX Jo et al. (2024) introduces formation-based exploration, which promotes visits to varied formations by instructing agents to thoroughly understand their current formations with the mutual information objective. CIA Liu et al. (2023) proposes distinguishing temporal credits of different agents by maximizing the mutual information between temporal credits and identity representations of agents to realize efficient credit assignment. The mutual information objective is further optimized by a contrastive learning lower bound. CTR Li et al. (2024) introduces a pre-training method that learns distinguishable trajectory representations using contrastive learning. TEE Li & Zhu maximizes a particle-based trajectory entropy estimator in a contrastive trajectory representation space to maximize the trajectory entropy of different agents. Inspired by these methods, to make the Wasserstein distance meaningful, we employ contrastive learning to distinguish trajectory representations of different agents. Moreover, different from the mutual information objective that does not necessarily encourage full exploration, we introduce a novel inner-product based Wasserstein distance to encourage efficient multi-agent diversity and exploration.

**Wasserstein Distance** The Wasserstein distance, recognized as an advanced measure of distribution dissimilarity, has attracted significant attention from the machine learning community. Numerous generative models (Arjovsky et al., 2017; Ambrogioni et al., 2018; Patrini et al., 2020; Tolstikhin et al.,

2018) have integrated the Wasserstein distance objective, demonstrating its effectiveness in scenarios where distributions degenerate onto sub-manifolds within pixel space. Recent advancements (Hu et al., 2024; Bettini et al., 2024) have leveraged the maximization of the Wasserstein distance to enhance policy differences among agents. MAPD (Hu et al., 2024) employs the Wasserstein distance as a metric for measuring policy differences by normalizing action distributions across different agents and computing the Wasserstein distance between them. In contrast to MAPD, our method constructs a contrastive trajectory representation space by contrasting the trajectories of different agents. This approach enables the learning of distinguishable trajectory representations, thereby making the Wasserstein distance more meaningful. Empirical results presented in Section 4 highlight the superiority of our proposed representation learning method over MAPD. DiCo (Bettini et al., 2024) introduces a metric based on the sum of Wasserstein distances to regulate diversity among agents, achieving favorable results in a simple multi-agent navigation task. However, it may encounter limitations in challenging multi-agent tasks due to the controlled diversity potentially leading to insufficient exploration. Moreover, these methods fall short in capturing the similarities between agent policies, resulting in an inefficient utilization of the Wasserstein distance, which ultimately hinders the performance of the proposed methods.

## C    Differences To Previous Mutual Information-Based Methods

Previous methods that focus on maximizing the mutual information between trajectories and agent identities generally formulate a variational intrinsic reward based on a variational lower bound Jiang & Lu (2021); Li et al. (2021); Charakorn et al. (2023); Jo et al. (2024):

$$r_v = \log q_\theta(i \mid \tau) - \log p(i), \tag{9}$$

Intuitively, the variational intrinsic reward $r_v$ motivates agents to explore distinct trajectories that the discriminator $q_\theta(i \mid \tau)$ can efficiently differentiate based on agent identities. However, this intrinsic reward $r_v$ does not measure the degree of difference between the trajectories. To address this limitation, our method introduces an intrinsic reward $r_w^a = \sum_{j=1, j \neq a}^{|A|} W(p_{\pi_a}, p_{\pi_j})^\top z_j$, which implies the Wasserstein distance between the trajectory distribution of the current agent and those of other agents in a latent representation space. By maximizing the intrinsic reward $r_w^a$, we enlarge the Wasserstein distance, thereby promoting greater trajectory diversity.

## D    Smoothed Wasserstein distance

In our method, we employ a smoothed version of the Wasserstein distance, denoted as $\tilde{W}_c(p, q)$. This variant introduces a regularization term to address potential issues with outliers or noise within the distributions, contributing to more robust optimization results Genevay et al. (2016). It is defined by the equation:

$$\tilde{W}_c(p, q) = \inf_{\gamma \in \Gamma[p,q]} \left[ \int_{\mathcal{X} \times \mathcal{Y}} c(x, y) d\gamma + \beta D_{KL}(\gamma \mid \xi) \right] \tag{10}$$

where $D_{KL}(\gamma \mid \xi)$ represents the KL divergence between the coupling $\gamma$ and $\xi$. $\xi$ is a reference measure over the product space $\mathcal{X} \times \mathcal{Y}$. The KL divergence serves as a regularizer added to the Wasserstein distance to smooth out the cost function. As $\beta \to 0$, the smoothed Wasserstein distance $\tilde{W}_c(p, q)$ converges to the Wasserstein distance $\mathcal{W}_c(p, q)$.

## E    The TD loss of QMIX

QMIX maximizes the agent team rewards by minimizing the TD loss to learn the optimal $Q_{tot}$ as follows:

$$\mathcal{L}_{TD} = \sum_{i=1}^{b} \left[ \left( r + \gamma \max_{\mathbf{u}_{t+1}} \bar{Q}_{tot}(s_{t+1}, \mathbf{u}_{t+1}) - Q_{tot}(s_t, \mathbf{u}_t) \right)^2 \right] \tag{11}$$

where $\bar{Q}_{tot}$ denotes the target network and $b$ represents the size of transition samples from the replay buffer $\mathcal{D}$. $r$ is the team reward shared among agents. It is important to note that since all agents' policies are jointly trained by minimizing the TD loss, we cannot directly apply each agent's intrinsic reward $r_w^a$ to the team reward $r$ to create a reward-shaping mechanism for independently training each agent's policy. Therefore, it is necessary to develop an additional intrinsic utility network to maximize the intrinsic reward $r_w^a$ as we do in our method.

# F  A THEORETICAL ANALYSIS OF OUR METHOD'S CONVERGENCE GUARANTEE

To address concerns that the exploration induced by the regularizer $\mathcal{L}_{TD}^w$ may affect MARL convergence, we provide a formal convergence guarantee. Our theoretical analysis proves the boundedness of our inner-product based intrinsic rewards. This condition is critical, as it ensures the diversity-seeking objective, driven by the intrinsic reward, does not overwhelm the primary team-reward objective. Without this bound, agents might adopt arbitrarily diverse policies at the expense of task performance. We next prove that our inner-product based Wasserstein distance, which serves as an intrinsic reward, is bounded. The proof relies on showing that each component of the reward formula is bounded and then applying the Cauchy-Schwarz inequality.

The proof follows by analyzing the components of the intrinsic reward formula and applying standard mathematical inequalities.

1. Deconstructing the Intrinsic Reward

    We define the inner-product based Wasserstein distance as an intrinsic reward $r_w^a$ for an agent $a$. The formula is given in Equation 6 as:

    $$r_w^a = \sum_{j=1, j \neq a}^{|A|} W(p_{\pi_a}, p_{\pi_j})^\top z_j$$

    To prove that $r_w^a$ is bounded, we must show that the two vectors in the inner product, $W(p_{\pi_a}, p_{\pi_j})$ and $z_j$, both have bounded norms.

2. Boundedness of the Components

    Latent Variable ($z_j$): The latent variable $z_j$ is randomly sampled from a fixed uniform distribution $p(z)$. A fixed uniform distribution has a defined, finite support. Therefore, any vector $z_j$ sampled from it will have a finite, bounded L2-norm. We can represent this bound as $||z_j||_2 \leq C_z$ for some constant $C_z > 0$.

    Wasserstein Distance ($W(p_{\pi_a}, p_{\pi_j})$): The Wasserstein distance is calculated between the distributions of trajectory representations ($c_t^a$) for different agents. The trajectory representations distribute around their corresponding identity representations that are uniformly distributed in the trajectory representation hypersphere. A hypersphere is a bounded space. This means the trajectory representations $c_t^a$ must have a bounded norm, which we can call $||c_t^a||_2 \leq C_{traj}$. The cost function used for the Wasserstein distance is the Euclidean distance, $c(c_t^1, c_t^2) = ||c_t^1 - c_t^2||_2$. Because the trajectory representations are confined to a bounded space, the Euclidean distance between any two of them is also bounded. By the triangle inequality, $||c_t^1 - c_t^2||_2 \leq ||c_t^1||_2 + ||c_t^2||_2 \leq 2C_{traj}$. Since the cost of transporting mass between any two points is bounded, the total Wasserstein distance, which represents the optimal transport cost, must also be bounded. We can denote this bound as $||W(p_{\pi_a}, p_{\pi_j})||_2 \leq C_W$ for some constant $C_W > 0$.

3. Applying the Cauchy-Schwarz Inequality

    Now that we've established that both vectors in the inner product are bounded, we can bound a single term from the summation using the Cauchy-Schwarz inequality, which states $|\langle u, v \rangle| \leq ||u||_2 \cdot ||v||_2$.

    For any single term $W(p_{\pi_a}, p_{\pi_j})^\top z_j$:

    $$|W(p_{\pi_a}, p_{\pi_j})^\top z_j| \leq ||W(p_{\pi_a}, p_{\pi_j})||_2 \cdot ||z_j||_2 \leq C_W \cdot C_z$$

    Each term in the summation is bounded by the constant product $C_W \cdot C_z$.

4. Bounding the Total Sum

Finally, we bound the entire intrinsic reward $r_w^a$ by applying the triangle inequality to the sum:

$$|r_w^a| = \left| \sum_{j=1, j \neq a}^{|A|} W(p_{\pi_a}, p_{\pi_j})^\top z_j \right| \leq \sum_{j=1, j \neq a}^{|A|} |W(p_{\pi_a}, p_{\pi_j})^\top z_j|$$

Substituting the bound for each term:

$$|r_w^a| \leq \sum_{j=1, j \neq a}^{|A|} (C_W \cdot C_z)$$

The number of terms in the sum is $(|A| - 1)$, where $|A|$ is the total number of agents. This leads to the final bound:

$$|r_w^a| \leq (|A| - 1) \cdot C_W \cdot C_z$$

Since the number of agents $|A|$ is finite and both $C_W$ and $C_z$ are finite constants, the inner-product based intrinsic reward $r_w^a$ is proven to be bounded.

## G  PSEUDOCODE FOR DMD

The pseudocode for DMD is given in Algorithm 1.

---

**Algorithm 1** Directional Metric-based Diversity (DMD)

---

Initialize identity representations $d^a$ and dual functions $\mu$ and $\nu$.
Initialize $Q_{tot}$ for QMIX.
**repeat**
    **for** *each episode* **do**
        Collect all agents' trajectories $\tau$ generated by the joint policy $\pi$.
        Store these trajectories in a replay buffer $D$.
    **end for**
    Randomly retrieve a batch of trajectories $\tau$ from the replay buffer $D$.
    Update the trajectory encoder $g_\theta$ by minimizing the CPC loss in Equation 3 to learn trajectory representations.
    Update dual functions $\mu$ and $\nu$ using SGD with the gradient in Equation 5.
    Calculate $r_w^a = \sum_{j=1, j \neq a}^{|A|} W(p_{\pi_a}, p_{\pi_j})^\top z_j$.
    Train each agent's policy $\pi_a$ by minimizing the total loss $\mathcal{L}_{total} = \mathcal{L}_{TD} + \alpha \mathcal{L}_{TD}^w$.
**until** $Q_{tot}$ *converges*

---

## H  THE IMPLEMENTATION OF DMD ON TOP OF POLICY-BASED METHODS

We have integrated our method, DMD, with the value-based algorithm QMIX and now extend this integration to policy-based methods. Specifically, we incorporate DMD into MAPPO, a state-of-the-art policy-based MARL algorithm as evaluated in SMAC. In MAPPO, all agents share an actor network and a critic network. Each agent learns its own critic, allowing us to easily add a shaped reward, $r_{env} + \alpha r_w$ (where $r_{env}$ is the environmental reward and $r_w$ is the Wasserstein distance-based intrinsic reward), to the reward-to-go $\hat{R}$ for updating each agent's critic. The other components of MAPPO remain unchanged. We also test the effectiveness of DMD+MAPPO in Pac-Men, SMAC, and SMACv2. The results, depicted in Table 1, demonstrate DMD+MAPPO's enhanced performance compared to baseline methods.

## I  ENVIRONMENTAL DETAILS AND ADDITIONAL EXPERIMENTAL RESULTS

### I.1  ENVIRONMENTAL DETAILS

In Pac-Men, four agents start in the central room of a maze. Each agent can only observe a 4×4 grid around them. Dots are randomly distributed in each edge room, and the agents aim to gather as

many dots as possible from these rooms. The lengths of the paths vary to evaluate the exploration of the environments, with path lengths set to 3, 6, 6, and 10 for the downward, leftward, rightward, and upward directions, respectively. Only one path is within the agent's observation range. Dots in each room respawn once all have been collected by the agents. The agents receive an environmental reward corresponding to the total number of dots consumed at each time step.

The SMAC benchmark comprises numerous cooperative tasks built upon Blizzard's real-time strategy game StarCraft II, aimed at evaluating the performance of different MARL algorithms. In SMAC, agent-level control leverages the Machine Learning APIs provided by StarCraft II and DeepMind's PySC2. Each task involves a combat scenario with two armies: one controlled by allied RL agents and the other by a non-learning game AI. The game ends when all units from any army are defeated or a predefined time limit is reached. The goal for the allied agents is to maximize the win rate by learning a sequence of behaviors to effectively collaborate in defeating enemy forces. An example of such collaboration is mastering kiting skills, where agents form strategic formations based on armor types to lure enemy units into pursuit while maintaining a safe distance to minimize damage. The SC2.4.10 version of StarCraft II is used in our work. Performance comparisons across different versions are infeasible. Experiments are conducted across six scenarios: 3s5z, 2c_vs_64zg, 7sz, 6h_vs_8z, corridor, and 3s5z_vs_3s6z.

SMAC is significantly limited by its lack of stochasticity. To address this issue, the newly released SMACv2 introduces modifications such as random team compositions and random start positions. These changes aim to introduce more stochastic elements into the environment to better evaluate the exploration capabilities of MARL algorithms. We conduct experiments in three SMACv2 scenarios: terran_5_vs_5, protoss_5_vs_5, and zerg_5_vs_5. In SMACv2, each race in StarCraft II uses three unit types. The probability of each unit type appearing in an episode remains fixed throughout the training and testing phases. Allied agents have the same unit types as their adversaries. In each episode, allied agents are randomly deployed on the map using either a reflect or surround style.

We provide the average returns of all algorithms in Pac-Men, SMAC, and SMACv2, and their standard deviation over five random seeds, in Table 1. The experimental results demonstrate the significant performance superiority of our method over baseline methods.

Table 1: Average returns of all algorithms in Pac-Men, SMAC, and SMACv2. $\pm$ denotes the standard deviation over five random seeds.

| Method | Pac-Men | SMAC | | | | | | SMACv2 | | |
|---|---|---|---|---|---|---|---|---|---|---|
| | | 3s5z | 2c_vs_64zg | 7sz | 6h_vs_8z | corridor | 3s5z_vs_3s6z | terran_5_vs_5 | protoss_5_vs_5 | zerg_5_vs_5 |
| QMIX | 0.21±0.04 | 0.72±0.13 | 0.85±0.08 | 0.17±0.02 | 0.23±0.03 | 0.57±0.07 | 0.36±0.12 | 0.68±0.03 | 0.53±0.05 | 0.41±0.04 |
| MAPPO | 0.49±0.03 | 0.81±0.05 | 0.83±0.04 | 0.52±0.06 | 0.53±0.03 | 0.62±0.05 | 0.57±0.08 | 0.52±0.04 | 0.47±0.03 | 0.37±0.03 |
| MAVEN | 0.32±0.06 | 0.51±0.21 | 0.72±0.06 | 0.00±0.00 | 0.42±0.04 | 0.36±0.08 | 0.18±0.15 | 0.58±0.04 | 0.31±0.05 | 0.29±0.03 |
| EOI | 0.41±0.05 | 0.87±0.07 | 0.83±0.02 | 0.37±0.03 | 0.08±0.03 | 0.25±0.11 | 0.42±0.13 | 0.65±0.05 | 0.42±0.03 | 0.47±0.04 |
| QTRAN | 0.28±0.08 | 0.21±0.19 | 0.75±0.05 | 0.00±0.00 | 0.02±0.02 | 0.08±0.07 | 0.02±0.01 | 0.42±0.02 | 0.40±0.04 | 0.25±0.02 |
| SCDS | 0.37±0.05 | 0.76±0.07 | 0.57±0.09 | 0.21±0.03 | 0.03±0.01 | 0.56±0.06 | 0.00±0.00 | 0.52±0.03 | 0.47±0.05 | 0.38±0.04 |
| PMIC | 0.34±0.03 | 0.82±0.03 | 0.79±0.05 | 0.58±0.02 | 0.51±0.05 | 0.37±0.03 | 0.18±0.06 | 0.47±0.03 | 0.36±0.02 | 0.42±0.02 |
| LIPO | 0.43±0.02 | 0.71±0.03 | 0.76±0.02 | 0.39±0.04 | 0.36±0.06 | 0.27±0.03 | 0.21±0.03 | 0.43±0.02 | 0.46±0.03 | 0.37±0.03 |
| FoX | 0.39±0.03 | 0.74±0.02 | 0.64±0.05 | 0.56±0.03 | 0.45±0.05 | 0.52±0.04 | 0.43±0.04 | 0.54±0.03 | 0.56±0.02 | 0.49±0.02 |
| RODE | 0.37±0.02 | 0.72±0.03 | 0.69±0.02 | 0.54±0.03 | 0.03±0.02 | 0.40±0.05 | 0.17±0.03 | 0.43±0.02 | 0.40±0.03 | 0.34±0.03 |
| MAPD | 0.23±0.03 | 0.63±0.04 | 0.65±0.03 | 0.04±0.02 | 0.26±0.07 | 0.12±0.08 | 0.10±0.06 | 0.44±0.03 | 0.38±0.02 | 0.36±0.03 |
| DiCo | 0.27±0.02 | 0.82±0.03 | 0.71±0.03 | 0.56±0.03 | 0.34±0.04 | 0.05±0.04 | 0.14±0.08 | 0.45±0.04 | 0.36±0.02 | 0.31±0.02 |
| DMD+QMIX | **0.89±0.02** | **0.94±0.02** | **0.97±0.03** | **0.92±0.02** | **0.91±0.03** | 0.88±0.04 | 0.86±0.05 | **0.93±0.04** | **0.89±0.03** | **0.86±0.03** |
| DMD+MAPPO | 0.86±0.02 | 0.93±0.03 | 0.90±0.04 | 0.87±0.02 | 0.83±0.05 | **0.92±0.04** | **0.89±0.04** | 0.92±0.02 | 0.87±0.04 | 0.85±0.03 |

## I.2 Additional results

**Google Research Football** We evaluate our method on three scenarios from Google Research Football (GRF), a complex, physics-based environment that simulates football gameplay. In this environment, agents must master strategic planning, coordination, and precise timing to succeed. The left-side players (excluding the goalkeeper) act as agents trained to develop cooperative policies, while the right-side players are controlled by the game engine. Each agent operates in a discrete action space with 19 options, including moving in eight directions, sliding, shooting, and passing. The observations available to each agent include the positions and movement directions of itself,

other agents, and the ball. As shown in Table 2, our method consistently outperforms the baseline methods in all scenarios.

Table 2: Performance comparisons of our method against the baseline methods in Google Research Football.

| Method | academy_3_vs_1_with_keeper | academy_4_vs_2_with_keeper | academy_counter_attack_hard |
|---|---|---|---|
| QMIX | 0.23±0.05 | 0.13±0.09 | 0.17±0.03 |
| MAPPO | 0.31±0.09 | 0.18±0.09 | 0.23±0.07 |
| MAVEN | 0.18±0.06 | 0.08±0.06 | 0.13±0.09 |
| EOI | 0.17±0.05 | 0.05±0.03 | 0.07±0.03 |
| QTRAN | 0.25±0.03 | 0.13±0.08 | 0.11±0.05 |
| SCDS | 0.42±0.13 | 0.25±0.11 | 0.47±0.06 |
| PMIC | 0.23±0.08 | 0.11±0.07 | 0.16±0.07 |
| LIPO | 0.19±0.05 | 0.07±0.03 | 0.12±0.05 |
| FoX | 0.57±0.05 | 0.41±0.13 | 0.33±0.08 |
| RODE | 0.37± 0.08 | 0.16±0.10 | 0.28±0.06 |
| MAPD | 0.23±0.11 | 0.11±0.06 | 0.19±0.07 |
| DiCo | 0.42 ±0.06 | 0.29±0.17 | 0.21±0.12 |
| DMD+QMIX | 0.83±0.11 | 0.78±0.07 | 0.73±0.13 |
| DMD+MAPPO | 0.80±0.06 | 0.75±0.09 | 0.69±0.07 |

**Stochasticity and Exploration** Although SMAC presents numerous challenging scenarios, there is a risk of agents overfitting to timesteps without responding to actual environmental states, due to the same team compositions and initial unit positions in each episode Ellis et al. (2022). To address this, we then evaluate our method on the SMACv2 Ellis et al. (2022) benchmark, which introduces stochasticity through random team compositions and random initial positions, compelling agents to persistently seek optimal policies. The performance comparisons depicted in Figure 6 show that our method consistently outperforms the baseline methods in all scenarios. Notably, our method enhances QMIX's effectiveness by incorporating the inner-product based Wasserstein distance objective as a regularizer to promote multi-agent diversity. Mutual information-based methods perform poorly, because the variational intrinsic rewards in these methods quickly converge once agents' trajectories are distinguished, limiting their ability to provide effective feedback for ongoing exploration. Additionally, MAPD falls short in promoting adequate exploration to adapt to environmental stochasticity, primarily due to the ineffectiveness of its Wasserstein distance-based incentives. In contrast, our method consistently offers efficient intrinsic rewards, effectively encouraging exploration. We present the visitation heatmaps in Figure 7, where agents trained with our method exhibit broader environmental exploration compared to those trained with baseline methods, which tend to concentrate only in specific areas.

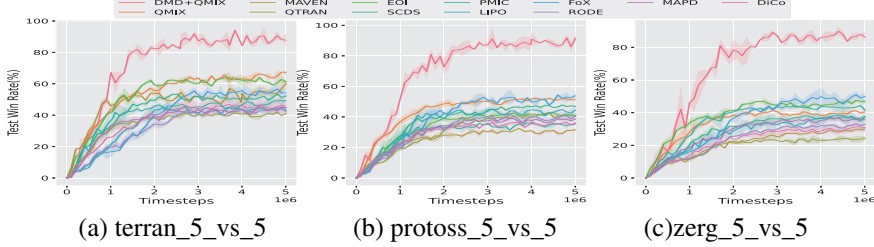

(a) terran_5_vs_5      (b) protoss_5_vs_5      (c)zerg_5_vs_5

Figure 6: Performance comparisons of DMD against baseline methods in the SMACv2 scenarios.

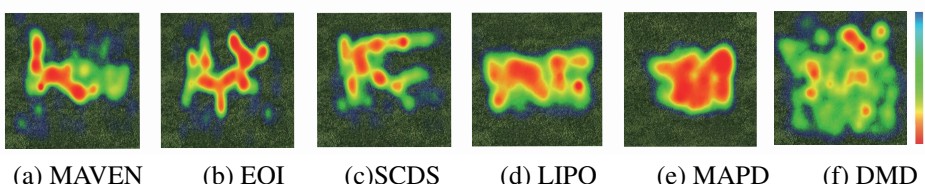

(a) MAVEN     (b) EOI     (c)SCDS     (d) LIPO     (e) MAPD     (f) DMD

Figure 7: Comparisons of visitation heatmaps between our proposed DMD and mutual information-based methods in the terran_5_vs_5 scenario.

**Homogeneous behaviors** Agents may sometimes need to adopt uniform behaviors. For example, in the SMAC scenarios, allied agents might simultaneously target and fire at the same enemy to quickly defeat it. To evaluate the learning efficiency of our method in scenarios requiring such homogeneous behaviors, we test it in four homogeneous SMAC scenarios that utilize the tactic of focus fire. The results, presented in Table 3, indicate that our method consistently outperforms QMIX across all these scenarios. This demonstrates that our method does not impede uniform behaviors when they are advantageous for achieving greater environmental rewards. Furthermore, our method promotes efficient exploration to develop such optimal cooperative behaviors.

Table 3: Performance comparison of our method against QMIX in homogeneous scenarios.

| Method | 8m | 5m_vs_6m | 8m_vs_9m | 10m_vs_11m |
|---|---|---|---|---|
| DMD+QMIX | 0.94±0.02 | 0.91±0.03 | 0.89±0.03 | 0.87± 0.04 |
| QMIX | 0.87±0.03 | 0.65±0.04 | 0.58±0.05 | 0.43±0.04 |

**Scalability** An increasing number of agents emerging in the environment challenges the scalability of MARL algorithms. The action space expands exponentially as the number of agents grows, emphasizing an urgent demand for exploration. In this section, we test the scalability of our method across four SMACv2 scenarios with varying numbers of agents: terran_5_vs_5, terran_10_vs_10, terran_15_vs_15, and terran_20_vs_20. The performance of our method and QMIX are depicted in Table 4. Our method essentially achieves better performance than QMIX in all tested scenarios. QMIX faces scalability challenges due to its inefficient exploration capabilities. Conversely, our method demonstrates robust scalability by effectively promoting extensive exploration of the action space. This is achieved by increasing the Wasserstein distance between different agents' trajectory distributions in the latent representation space, ensuring efficient exploration.

Table 4: Performance comparison of our method against QMIX in SMACv2 scenarios with increasing numbers of agents

| Method | terran_5_vs_5 | terran_10_vs_10 | terran_15_vs_15 | terran_20_vs_20 |
|---|---|---|---|---|
| DMD+QMIX | 0.93±0.04 | 0.92 ±0.03 | 0.90 ±0.05 | 0.87 ±0.04 |
| QMIX | 0.68±0.03 | 0.39±0.04 | 0.24 ±0.06 | 0.11±0.05 |

Moreover, we evaluated our method on a large-scale multi-agent benchmark, Magent. The Magent platform supports large-scale multi-agent reinforcement learning with tasks such as pursuit, battle, combined arms, and tiger deer. We tested our method on the battle task with varying numbers of agents. The evaluation results are shown in the table 5. Compared to QMIX, our method continues to outperform it and scales well as the number of agents increases.

Table 5: Scalability evaluations on a large-scale multi-agent benchmark

| Number of agents | QMIX | DMD+QMIX |
|---|---|---|
| 50 | 376±59 | 273±73 |
| 75 | 892±228 | 482±183 |
| 100 | 2759±695 | 1196±373 |
| 125 | 3269±1427 | 2375±562 |

## J  COMPARISON WITH $\epsilon$-GREEDY

The $\epsilon$-greedy method is a widely adopted exploration strategy in many RL algorithms, where increasing the value of $\epsilon$ generally promotes greater exploration. In this section, we compare our Wasserstein distance-based method with $\epsilon$-greedy to demonstrate its effectiveness in enhancing exploration within MARL. For this comparison, we set $\epsilon$ to 0.05, 0.075, and 0.1 for QMIX and evaluate these settings in challenging scenarios, including corridor, 3s5z_vs_3s6z, terran_5_vs_5, and protoss_5_vs_5. As shown in Table 6, our entropy maximization method significantly outperforms the $\epsilon$-greedy approach in fostering exploration. Notably, increasing $\epsilon$ does not lead to substantial performance improvements. In multi-agent settings, higher $\epsilon$ values primarily introduce more randomness in individual agents' action selections without effectively enhancing diversity or coordination among agents, as they do not account for the trajectories of other agents, resulting in suboptimal exploration.

Table 6: Comparison of performance between our method and QMIX using various $\epsilon$ values

| Method | corridor | 3s5z_vs_3s6z | terran_5_vs_5 | protoss_5_vs_5 |
|---|---|---|---|---|
| $\epsilon = 0.05$ (QMIX) | 0.57 ±0.07 | 0.36 ±0.12 | 0.68 ±0.03 | 0.53 ±0.05 |
| $\epsilon = 0.075$ (QMIX) | 0.61 ±0.04 | 0.39 ±0.11 | 0.72 ±0.04 | 0.62 ±0.07 |
| $\epsilon = 0.1$ (QMIX) | 0.63 ±0.06 | 0.44 ± 0.15 | 0.74 ±0.03 | 0.69 ±0.06 |
| Wasserstein distance (our method) | 0.88 ± 0.04 | 0.86 ± 0.05 | 0.93 ±0.04 | 0.89 ±0.03 |

## K    EVALUATIONS OF DIFFERENT KERNEL FUNCTIONS

In our paper, we use the Gaussian kernel by default. Alternatively, a linear kernel can be employed to parameterize dual functions. To assess the effectiveness of using a linear kernel, we design a linear kernel variant and evaluate it in the super hard scenarios of SMAC. The results, presented in Table 7, indicate a significant performance decline when using the linear kernel for dual functions. We suspect that this is due to the dual function potentially not being linear, causing the linear kernel to limit the representational capacity of the dual function.

Table 7: Performance comparisons of DMD with different kernel functions in the scenarios of SMAC

| Method | 6h_vs_8z | corridor | 3s5z_vs_3s6z |
|---|---|---|---|
| DMD (Linear Kernel) | 0.43± 0.04 | 0.37 ± 0.07 | 0.32 ± 0.06 |
| DMD (Ours) | 0.91 ± 0.03 | 0.88 ± 0.04 | 0.86 ± 0.05 |

## L    EVALUATIONS OF DIFFERENT VALUES FOR THE WEIGHT OF THE INTRINSIC REWARD $\alpha$

Table 10 presents the values of the intrinsic reward weight $\alpha$ used in different scenarios. To examine the impact of varying weights for intrinsic rewards, we conduct experiments with different $\alpha$ values in the easy scenario 3s5z and the super hard scenario corridor. The results, shown in Table 8, indicate that our method exhibits low sensitivity to the choice of $\alpha$. Even with sub-optimal weights, the performance does not significantly degrade, including in the super hard scenario.

Table 8: Performance comparisons of DMD with different values for the weight of the intrinsic reward $\alpha$.

| Method | 3s5z | | | corridor | | |
|---|---|---|---|---|---|---|
| | $\alpha = 0.02$ | $\alpha = 0.05$ | $\alpha = 0.1$ | $\alpha = 0.02$ | $\alpha = 0.05$ | $\alpha = 0.1$ |
| DMD | 0.87 ± 0.03 | 0.85 ± 0.05 | 0.90 ± 0.02 | 0.85 ± 0.06 | 0.88 ± 0.04 | 0.86 ± 0.03 |

## M    EVALUATIONS OF DIFFERENT COST FUNCTIONS

In our paper, we primarily use the Wasserstein distance to promote sufficient exploration, adopting the Euclidean distance as the cost function, similar to many previous works. Alternatively, cosine similarity can be used as the cost function to capture directional differences between data points. To evaluate this approach, we tested the cosine similarity in the Pac-Men scenario, where agents are required to move in different directions. The results, presented in Table 9, show that the Wasserstein distance based on cosine similarity yields higher rewards in Pac-Men. We chose to use the default Euclidean distance in our experiments to maintain consistency with prior works employing the Wasserstein distance, ensuring a fair comparison.

## N    TRAINING DETAILS AND HYPERPARAMETERS

For consistency and fairness in comparison, we use the same common hyperparameters and policy network architecture for all tested methods, with specifics listed in Table 10. To implement the trajectory encoder, we utilize a two-layer MLP with a hidden size of 64 for the encoder $g_{\theta_e}$, which

Table 9: Performance comparisons of DMD using different cost functions.

| Method | Pac-Men |
|---|---|
| DMD (Cosine Similarity) | $92 \pm 0.03$ |
| DMD (Euclidean Distance) | $89 \pm 0.02$ |

includes batch normalization, and a GRU for the autoregressive model $g_{\theta_g}$. The identity representation is represented by a randomly initialized vector with a dimension of 64 that has the same dimension with the trajectory representation. The dual vector used to parameterize the dual function has a dimension of 64. For integration with QMIX, we use a two-layer MLP with a hidden size of 64 for the intrinsic utility network, while retaining the same additional components as in QMIX.

The policy networks for all agents are based on Deep Recurrent Q-Networks. Concretely, each agent's policy network receives a local observation as input at each time step, processes it through a fully-connected hidden layer, then a GRU unit, and finally through a fully-connected layer that produces $U$ outputs, where $U$ represents the number of possible actions. To facilitate faster training, all agents share the same policy network parameters. For target network updates in SMAC and SMACv2, we employ hard updates every 200 episodes. In Pac-Men, soft updates are used with a momentum of 0.01 for updating target networks. The evaluation interval is set to 10,000 steps, followed by 32 test episodes. All methods are run for 5 million steps. The size of the replay buffer is maintained at 5K. Our method is implemented using NumPy and PyTorch, and all experiments are conducted on a NVIDIA GeForce RTX 4090 GPU.

Table 10: Hyperparameters

| | Pac-Men | SMAC | SMACv2 |
|---|---|---|---|
| hidden dimension | 64 | 128 | |
| learning rate | 0.0003 | 0.005 | |
| optimizer | | Adam | |
| target update | 0.01(soft) | 200(hard) | |
| batch size | 32 | 64 | |
| $\beta$ | 0.03 | 0.05 | |
| $\alpha$ for DMD+QMIX | 0.01 | 0.005 for 3s5z, 2c_vs_64zg, 0.05 for 7sz, 6h_vs_8z, corridor, and 3s5z_vs_3s6z | 0.03 |
| $\alpha$ for DMD+MAPPO | 0.01 | 0.005 for 3s5z, 2c_vs_64zg, 0.03 for 7sz, 6h_vs_8z, corridor, and 3s5z_vs_3s6z | 0.03 |
| epsilon anneal time | 200,000 | 200,000 for 3s5z, 2c_vs_64zg, 500,000 for 7sz, 6h_vs_8z, corridor, and 3s5z_vs_3s6z | 500,000 |

## O  VISUALIZATIONS

Challenging tasks often require complex cooperative behaviors, necessitating that agents learn diverse policies. We further showcase some visualization examples of these diverse policies learned by our method in super hard scenarios (6h_vs_8z, corridor, and 3s5z_vs_3s6z) in Figure 8. In the 6h_vs_8z scenario, one agent initially separates from the group, prompting most enemies to track this lone agent's movements. This agent continues to move away, drawing enemy fire and providing cover for the teammates. Meanwhile, the other agents take advantage of this diversion to quickly encircle and overwhelm the few remaining enemies. Such tactics help break up the enemy's concentrated attacks, demonstrating the effectiveness of our method in encouraging multi-agent diversity and enhancing cooperative behaviors. Similar strategies are evident in the other two scenarios. If all agents were to approach the enemies directly, they would likely suffer immediate defeat.

## P  COMPUTATIONAL COST ANALYSIS

We note that compared to QMIX, our method needs to train the trajectory encoder and the dual functions to calculate the Wasserstein distance objective, which introduces additional computational overhead. We then present a comparison of training time between QMIX and our method in the three

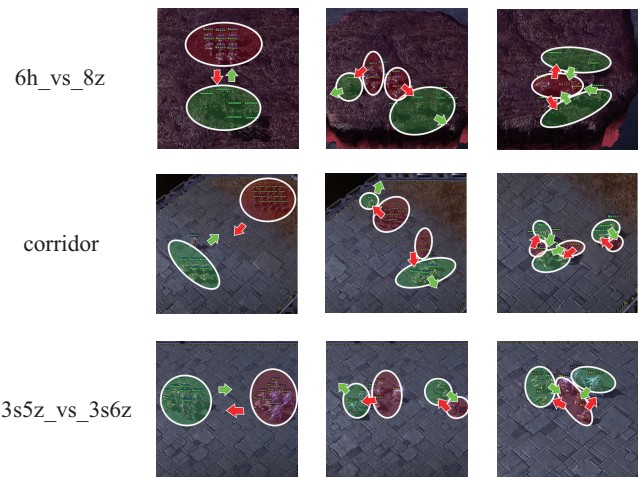

6h_vs_8z

corridor

3s5z_vs_3s6z

Figure 8: Visualizations of diverse policies evolving in the scenarios 6h_vs_8z (top), corridor (medium), and 3s5z_vs_3s6z (bottom), from the initial stages (left) to the final stages (right). Green and red shadows indicate the positions of agents and enemies, respectively. Similarly, green and red arrows depict the movement directions of the agents and enemies, respectively.

super hard scenarios of SMAC on the same computing platform in the table 11. The results show that, compared to QMIX, our proposed method does not consume significant extra training time.

Table 11: Comparisons of training time between QMIX and our method

| Methods | 6h_vs_8z | corridor | 3s5z_vs_3s6z |
|---|---|---|---|
| QMIX | 8h 35m 29s | 7h 17m 30s | 10h 42m 18s |
| DMD+QMIX | 8h 47m 36s | 7h 26m 42s | 10h 50m 39s |

## Q  PERFORMANCE COMPARISONS OF OUR METHOD AGAINST AND CIA, CTR, AND TEE

We compare the performance of our method against CIA Liu et al. (2023), CTR Li et al. (2024), and TEE Li & Zhu in the SMAC-Exp benchmark, which is designed to evaluate the exploration capability of MARL algorithms in efficiently learning implicit multi-stage tasks, environmental factors, and micro-control. We examine our method in three hard tasks: Off_complicated, Off_hard, and Off_superhard. The experimental results are shown in Table 12. We note that our method maintains superior performance across all three tasks. Although these works are all based on contrastive trajectory representations, our proposed method using the inner-product-based intrinsic reward achieves more robust performance.

Table 12: Performance comparisons of our method against and CIA, CTR, and TEE

| Methods | Off_complicated | Off_hard | Off_superhard |
|---|---|---|---|
| CIA | 0.13±0.07 | 0.47±0.11 | 0.03±0.02 |
| CTR | 0.42±0.19 | 0.32±0.06 | 0.12±0.07 |
| TEE | 0.64±0.16 | 0.56±0.09 | 0.06±0.04 |
| DMD+QMIX | 0.78±0.13 | 0.72±0.05 | 0.37±0.18 |

## R  LIMITATIONS AND FUTURE DIRECTIONS

The cost function of the Wasserstein distance determines how the probability mass is transferred. For simplicity, we resort to the Euclidean distance as the cost function of the Wasserstein distance in the

experimental settings. However, selecting an optimal cost function for the Wasserstein distance to address specific multi-agent tasks still remains a challenge. Therefore, adapting the cost function used in our method to efficiently tackle a wide range of multi-agent tasks is a key objective for our future research.

## S   PARAMETER SHARING VS. NON-PARAMETER SHARING

In our experiments, all baselines share the same policy network parameters, as discussed in the training details section. We focus on the parameter-sharing setting because our method is built to address its inherent limitations. We next compare the performance of our method with baselines under non-parameter-sharing settings. The results are shown in Table 13. Compared to baselines that learn trajectory discriminators (EOI) or solely maximize Wasserstein distance (MAPD), our method achieves better performance.

Table 13: Performance comparisons of our methd against baselines under non-parameter sharing setting

| Methods | 6h_vs_8z | corridor | 3s5z_vs_3s6z |
|---------|----------|----------|--------------|
| QMIX | 0.17±0.05 | 0.39±0.12 | 0.06±0.03 |
| EOI | 0.00±0.00 | 0.09±0.04 | 0.13±0.08 |
| MAPD | 0.05±0.03 | 0.07±0.05 | 0.00±0.00 |
| DMD+QMIX | 0.68±0.19 | 0.71±0.07 | 0.75±0.12 |

