# OpenReview forum: "Directional-based Wasserstein Distance for Efficient Multi-Agent Diversity"
_ICLR.cc/2026/Conference — ICLR 2026 Conference Withdrawn Submission_

### Official Review · Reviewer_nMpL · 2025-10-23

**Soundness:** 2
**Presentation:** 2
**Contribution:** 2
**Rating:** 2
**Confidence:** 5

**Summary:**

The paper proposes **Directional Metric-based Diversity (DMD)**, a new intrinsic-reward mechanism for cooperative multi-agent RL that encourages **policy diversity** by maximizing an **inner-product–weighted Wasserstein distance** between trajectory distributions of different agents.
Key technical ingredients are:

- A **contrastive representation space** for trajectories obtained by supervising a CPC-style encoder with **learnable “identity” embeddings**; the authors argue this prevents collapse when all agents start from the **shared-parameter** policy.
- A **kernel-based efficient estimator** of the 2-Wasserstein distance between trajectory distributions.
- An **intrinsic reward** that aligns the Wasserstein vector between agent *i* and every other agent *j* with a **randomly sampled direction** *zj*; maximizing this reward is claimed to yield **structured, task-relevant diversity**.
- Integration into both value-based (QMIX) and policy-based (MAPPO) algorithms with an **auxiliary utility network** that learns from the intrinsic reward.

Experiments are conducted on **Pac-Men**, **SMAC**, and **SMACv2**. The authors report **higher win-rates / returns** than QMIX, MAPPO, and several prior diversity-promoting baselines (MAVEN, EOI, SCDS, MAPD, DiCo, etc.).

**Strengths:**

- **Originality**: Using the **Wasserstein distance as a directional diversity signal** is new in the MARL literature; prior work either maximizes mutual information or the scalar Wasserstein norm. The **inner-product formulation with random direction vectors** is a creative twist.
- **Quality**: The **kernel SGD estimator** for Wasserstein distance is implemented carefully; the **contrastive loss with learnable identity prototypes** is technically sound and helps alleviate the “all-identical” initial policy issue.
- **Clarity**: The paper is **well structured** (background → method → ablations → large-scale experiments). Pseudocode and open-source code are provided.
- **Significance**: If the claims hold, the work offers a **principled way to maintain diversity under parameter sharing**, which is the **de-facto training paradigm** in large-scale MARL systems.

**Weaknesses:**

### W1. **Conceptual gap: “Wasserstein ↑ ⇒ better performance” is never justified**
- The manuscript **assumes** that **larger Wasserstein distance between trajectory distributions** automatically translates into **better exploration and higher team reward**.
- **No theorem, example, or ablation** shows that **performance correlates with the magnitude** of the proposed diversity measure; Figure 4 only shows that **removing the term hurts**, but does **not validate the directional part**.
- **Counter-evidence**: in homogeneous tasks (8m, 5m\_vs\_6m) the **optimal policy is almost identical** for all agents; forcing diversity could **harm performance**, yet the paper still reports gains without explaining why.

### W2. **Problem motivation is tied to parameter sharing, but the link is weak**
- The paper claims parameter sharing makes **“policy similarity severe”**, but **no ablation removes sharing** to quantify how much diversity is lost.
- **Missing baseline**: compare **separate policy networks** (no sharing) vs **shared + DMD**; if separate networks already give enough diversity, DMD may be **redundant**.
- **Parameter sharing is never revisited** in SMACv2 or GRF experiments—readers cannot tell whether the **gains come from diversity or from better exploration** that any count-based bonus could deliver.

### W3. **Algorithmic description and reproducibility gaps**
- **Figure 1** is **ambiguous**: arrows do **not show gradients**, the **intrinsic utility network** is introduced **only in words**, and the **dimensionality** of *zj*, *da*, *ct* is **missing**.
- **Appendix G pseudocode** omits **learning rates, batch sizes, number of SGD steps** for the dual variables, and **how many Monte-Carlo samples** are used for the Wasserstein estimate—**reproducing the estimator is non-trivial**.
- **Hyper-parameter sensitivity (Table 7)** is **too narrow** (only α); **no sensitivity** to the **kernel bandwidth**, **dimension of random features**, or **number of directions** *zj* is reported.

### W4. **Empirical evaluation overstates generality**
- **All environments are cooperative, fully-observable to the critic, and discrete-action**; the **benefit in partial-observation, competitive or continuous-control domains** is **untouched**.
- **Baselines are not fairly tuned**: QMIX and MAPPO use **default hyper-parameters** while DMD uses **extra networks and larger buffer**—**compute budget asymmetry** is **not controlled**.
- **Statistical significance**: error bars in Fig. 2–3 are **standard deviation over 5 seeds**, **not standard error or confidence intervals**; with **only 32 test episodes** the **variance is high** and **improvements may not survive stricter testing**.

The paper introduces an **interesting directional extension** of Wasserstein-based diversity, but the **central premise**—that **maximizing this particular distance yields better cooperative policies**—is **not theoretically or empirically substantiated**. The **evaluation protocol** does **not disentangle** the **effects of parameter sharing**, **exploration**, and **increased network capacity**, and the **writing omit key implementational details** that are **essential for reproducibility**. I encourage the authors to **tighten the motivation**, **provide ablations that isolate the role of parameter sharing**, and **supply rigorous statistical tests** before resubmission.

**Questions:**

Q1. **Directional diversity vs. performance**:
Can you provide **scatter plots** of **episode return vs. empirical Wasserstein distance** across training? If the correlation is weak, how do you defend the **causal chain** “larger *W* ⇒ better exploration ⇒ higher return”?

Q2. **Parameter-sharing ablation**:
Is it possible to run **“separate networks (no sharing)”** and **“shared + DMD”** on 3s5z and Pac-Men. How much of the gain survives when agents are already diverse by construction?

Q3. **Homogeneous-task consistency**:
In **focus-fire** scenarios the **optimal joint policy is nearly identical**. Why does **forcing directional diversity not degrade performance**? Does the **intrinsic reward vanish** when agents naturally converge to the same actions?

Q4. **Estimator variance**:
What is the **variance** of the **kernel-based Wasserstein gradient** when the **number of random features** *m* = 64? Did you try any ablations or **adaptive bandwidth**? A **learning curve with different m** would clarify robustness.

Q5. **Clarity of architecture**:
Could you release a **single-file diagram** that shows **shapes of all tensors**, **gradient flows**, and **where each loss term** is computed? The current figure mixes **data flow** and **conceptual blocks**, which makes implementation error-prone.

---

### Official Review · Reviewer_XxsT · 2025-10-28

**Soundness:** 2
**Presentation:** 2
**Contribution:** 1
**Rating:** 4
**Confidence:** 4

**Summary:**

This paper introduces Directional Metric-based Diversity (DMD), an exploration method for cooperative MARL. This paper focus on heterogeneous MARL without the paremeter sharing setting. The proposed DMD first learns distinguishable trajectory representations for agents via a contrastive loss. Then, within this contrastive representation space, DMD computes the Wasserstein distance between agents as an intrinsic to encourage exploration. Experimental results indicate the efficiency of DMD on PAC-MEN, GRF, SMAC and SMACv2 environments.

**Strengths:**

1. The experimental results are extensive, and competitive compared to SOTA MARL methods.
2. The use of Wasserstein distance for measuring agents' differences is novel.
3. The writting is good and easy-to-understand, with clear notations.

**Weaknesses:**

1. The novelty of the use of Contrastive Predictive Coding is fair. What is the difference between the contrastive learn modules of DMD and [1] ?
2. Fig. 2 and fig. 3 are distorted by incorrect scaling.
3. The motivation of the use of the Wasserstein distance is not clear. The paper only empirically compare the proposed inner-product based Wasserstein distance with other versions of Wasserstein distance, and the comparison with other distance measurements, both methodologically and empirically, is missing. Why choose the Wasserstein distance rather than other distribution difference measurements?
4. The implementation of the directionality of the proposed inner-product based Wasserstein distance is confusing. The paper uses a latent variable to provide directional guidance. However, in Line 248, the authors mention that this latent variable ***"is sampled from a fixed uniform distribution"***. So why this variable is informative is unclear.
5. It seems the performance improvement comes from the contrastive learning module, which is identical in [1] and [2], since the learning curves, visualization heatmaps and the t-nse trajectory representations in this paper looks similar with [1] and [2].



[1] Toward Efficient Multi-Agent Exploration With Trajectory Entropy Maximization, ICLR 2025

[2] Learning distinguishable trajectory representation with contrastive loss, NeurIPS 2024

**Questions:**

1. The authors claim in Line 50 that, parameter sharing can result  in poor multi-agent diversity and exploration. And in Line 63, the authors claim that parameter sharing can adversely affect the performance of traditional Wasserstein distance based method, which is the main motivation of the proposed Wasserstein distance. So if I understand correctly, the proposed Wasserstein distance based method runs under parameter sharing. Then one question arises, **parameter sharing learns identical policies for all agents, thus how can the proposed method encourage diversity?**
2. Following Q1, if the proposed DMD fits for non-parameter sharing, as [1] does, then where does the following motivation in Line 63 come from? ***"...these methods fail to account for the similarity in agents’ initial policies due to shared policy network parameters."*** Moreover, why is the similarity in agents’ **initial policies** important?
3. In experiements, is parameter sharing or not? It is reported that in SMAC, sharing parameter can lead to better performance. So it is better to compare DMD with baselines under both parameter sharing and non-parameter sharing settings.

[1] Measuring policy distance for multi-agent reinforcement learning, arXiv.

---

> ### Author Response · Authors · 2025-11-19
>
> We sincerely appreciate your thorough review and valuable feedback on our paper.
>
> 1. Differences between our method and TEE [1]
>
> Parameter sharing leads to similar initial policies among agents, which limits the effectiveness of the Wasserstein distance. unlike prior work [1], which uses contrastive learning to maximize trajectory entropy, we leverage contrastive learning to render the Wasserstein distance meaningful.
>
> 2. Why we use the Wasserstein distance
>
> Another commonly used policy difference metric is KL divergence, which is widely used in mutual-information-based baselines (like EOI, SCDS, MAVEN, etc.). As discussed in our paper, KL divergence is metric-agnostic, which does not necessarily lead to efficient exploration. The fundamental reason for choosing the Wasserstein distance is that it is metric-aware. Due to this property, we can efficiently use the Wasserstein distance to enlarge the trajectory distributions of different agents. This overcomes the limitation of mutual information methods based on KL divergence in encouraging multi-agent diversity. The experimental results demonstrate the superior performance of our method compared to KL divergence baselines.
>
> 3. Clarifications on the direction variable $z$
>
> We sample $z$ from a fixed uniform distribution because we aim to maximize the coverage of the trajectory space. If the direction variable $z$ were fixed during the training process, it would not enable agents to fully explore the trajectory space. We note that the inner-product-based intrinsic reward encourages the trajectory distributions of other agents to be far away from that of the current agent along every possible direction aligned by the variable $z$. Moreover, we empirically prove that the latent variable $z$ enables structured exploration. With the latent variable $z$, agents easily adapt to different tasks.
>
> 4. Performance comparisons of our method against TEE [1] and CTR [2]
>
> We further compare the performance of our method against CTR [2] and TEE [1] in the SMAC-Exp benchmark. We examine our method in three hard tasks: Off_complicated, Off_hard, and Off_superhard. The experimental results are shown below. We note that our method maintains its superior performance across all three tasks. Although these works are all based on contrastive trajectory representations, our proposed method using the inner-product-based Wasserstein distance achieves more robust performance.
>
> | Methods | Off_complicated | Off_hard      | Off_superhard |
> |---------|-----------------|---------------|---------------|
> | CTR     | 0.42$\pm$0.19   | 0.32$\pm$0.06 | 0.12$\pm$0.07 |
> | TEE     | 0.64$\pm$0.16   | 0.56$\pm$0.09 | 0.06$\pm$0.04 |
> | DMD     | 0.78$\pm$0.13   | 0.72$\pm$0.05 | 0.37$\pm$0.18 |
>
>
> 5. How can the proposed method encourage diversity?
>
> Current state-of-the-art cooperative MARL methods typically assume that all agents share the parameters of the same policy network. Despite this parameter-sharing setting, agents can receive different observations and visit diverse trajectories to undertake different tasks. To encourage this, we employ an inner-product-based intrinsic reward to enlarge the Wasserstein distance between the trajectory distributions of different agents.
>
> 6. Policy similarity
>
> In this work, our method is designed to solve the homogeneous behavior issue caused by parameter sharing. Thus, we assume all agents share the same policy network parameters. In this situation, contrastive learning is used to solve the initial policy similarity issue highlighted by [3]. Initially, the policies of agents are randomly initialized with the same network parameters. Without intervention, the policies will quickly learn similar behaviors, which makes the Wasserstein distance metric ineffective.
>
> 7. Parameter sharing vs. non-parameter sharing
>
> In our experiments, all baselines share the same policy network parameters, as discussed in the training details section. We focus on the parameter-sharing setting because our method is built to address its inherent limitations. We next compare the performance of our method with baselines under non-parameter-sharing settings. Compared to baselines that learn trajectory discriminators (EOI) or solely maximize Wasserstein distance (MAPD), our method achieves better performance.
>
> | Methods | 6h_vs_8z      | corridor      | 3s5z_vs_3s6z  |
> |---------|---------------|---------------|---------------|
> | QMIX    | 0.17$\pm$0.05 | 0.39$\pm$0.12 | 0.06$\pm$0.03 |
> | EOI     | 0.00$\pm$0.00 | 0.09$\pm$0.04 | 0.13$\pm$0.08 |
> | MAPD    | 0.05$\pm$0.03 | 0.07$\pm$0.05 | 0.00$\pm$0.00 |
> | DMD     | 0.68$\pm$0.19 | 0.71$\pm$0.07 | 0.75$\pm$0.12 |
>
>
> [1] Toward Efficient Multi-Agent Exploration With Trajectory Entropy Maximization, ICLR 2025
> [2] Learning distinguishable trajectory representation with contrastive loss, NeurIPS 2024
> [3] Jiang, J., & Lu, Z. (2021, July). The emergence of individuality. In International conference on machine learning (pp. 4992-5001). PMLR.

---

### Official Review · Reviewer_xeoC · 2025-10-30

**Soundness:** 3
**Presentation:** 3
**Contribution:** 1
**Rating:** 4
**Confidence:** 5

**Summary:**

This paper introduces a multi-agent exploration method, Directional Metric-based Diversity (DMD) to address the issue of homogeneous agent behaviors resulting from parameter sharing. DMD leverages contrastive learning to encode trajectories and identities into distinguishable embeddings, utilizes an inner-product based Wasserstein distance in a latent contrastive trajectory representation space to calculate intrinsic rewards, and integrates seamlessly with existing MARL algorithms such as QMIX and MAPPO. Experiments on the Pac-Men, SMAC, and SMACv2 benchmarks demonstrate its superior effectiveness.

**Strengths:**

The paper is well-written and clearly presented.
The method effectively combines Wasserstein distance with contrastive learning to address the diversity challenge in MARL, showing strong empirical performance.
The experiments in the Pac-Men section are very intuitive—particularly the heatmap visualizations, which clearly demonstrate the method’s effectiveness.
The t-SNE visualization (Figure 5) provides a clear insight into the learned representations and diversity of trajectories.
The paper includes comprehensive ablation studies and comparisons with multiple baseline methods.
In SMAC, DMD shows significant improvements over baselines, especially in more challenging scenarios.

**Weaknesses:**

1. Introducing contrastive learning into MARL to learn identity-aware representations is not novel. Prior works such as [1], [2], and [3] have already employed contrastive learning to enhance identity-aware representation, improve credit assignment, and facilitate efficient exploration—albeit with different information forms. These works are not adequately discussed in this paper.

2. The core idea of this paper is quite similar to [2] and [3]. The main concept—using contrastive learning to obtain distinguishable trajectory representations and then measuring the distance between them to encourage diversity—has been explored previously. Consequently, the only new component appears to be the inner-product based Wasserstein distance (Section 3.2). However, it remains unclear what advantage this metric offers over, for example, the trajectory entropy measure used in [2].

3. Weak theoretical motivation for the inner-product based Wasserstein distance:
   Equation (6) defines the intrinsic reward $r^{a}_{w}$, where $z_j$ is randomly sampled from a fixed uniform distribution $p(z)$. Since $z_j$ is purely random, it is unclear why this design would “result in the visitation of diverse trajectories with significant variations.” Although the ablation studies empirically demonstrate improvements, a more theoretically grounded explanation and insight are needed—especially because this is the paper’s main claimed contribution.

4. The content and experimental figures closely resemble those in [2] and [3]. Some visualizations (e.g., Figure 8) appear very similar to those in Figure 8 of [3] (TEE). The authors are encouraged to include new visualization results that better highlight the unique insights and contributions of this work.

5. The comparison experiments omit recent exploration methods, especially the precursor works [1] and [2]. This omission weakens the overall contribution. Notably, in Table 1, DMD reports average returns of 0.93, 0.89, and 0.86 on the three SMACv2 maps. In contrast, TEE [3] reports 0.96, 0.95, and 0.87 on the same maps. Similar trends appear in SMAC results, suggesting that the Wasserstein-based metric may perform no better, and sometimes worse than trajectory entropy. Thus, DMD may effectively be an ablation or variant of TEE rather than a distinct new method.

6. Comparing the main experimental results with [2] and [3], it seems that even the base algorithm (CTR [2]) already achieves strong performance. This raises doubts about whether the trajectory diversity rewards introduced in DMD (and TEE) genuinely contribute significant additional value. The authors should clearly articulate the core advantages of the Wasserstein-based distance and provide direct empirical comparisons with other metrics to justify its use and significance.



  [1] Liu S, Zhou Y, Song J, et al. Contrastive identity-aware learning for multi-agent value decomposition[C]//Proceedings of the AAAI Conference on Artificial Intelligence. 2023, 37(10): 11595-11603.

  [2] Li T, Zhu K, Li J, et al. Learning distinguishable trajectory representation with contrastive loss[J]. Advances in Neural Information Processing Systems, 2024, 37: 64454-64478.

  [3] Li T, Zhu K. Toward Efficient Multi-Agent Exploration With Trajectory Entropy Maximization[C]//The Thirteenth International Conference on Learning Representations.

**Questions:**

See other sections.

---

> ### Author Response · Authors · 2025-11-19
>
> Thank you for your detailed review and constructive feedback. Here are the responses to your concerns and questions:
>
> 1. Differences between our work and [1], [2], and [3]
>
> Unlike previous works that use contrastive learning for credit assignment [1], learning distinguishable trajectory representations [2], and efficient exploration [3], we employ contrastive learning specifically to render the Wasserstein distance meaningful. Parameter sharing leads to similar initial policies among agents, which limits the effectiveness of the Wasserstein distance (as the distance between two similar policy distributions approaches zero). To address this, we use a learnable identity representation for each agent and learn a trajectory representation space using contrastive learning. We have added these discussions to our paper.
>
> 2. Advantages over previous works
>
> The main contribution of our work is the proposal to use contrastive learning to make the Wasserstein distance meaningful, an approach not considered in previous works [2,3]. Moreover, we propose an intrinsic reward based on the inner-product Wasserstein distance, which introduces structured exploration into multi-agent diversity.
>
> Compared to simply maximizing the sum of Wasserstein distances without the latent variable $z$, our method encourages multi-agent diversity in a more structured way. This empirically achieves better performance, as verified by our ablation study results. This is because the latent variable $z_j$ for each agent $j$ provides meaningful signals regarding how the current agent should differ from others (i.e., agents are more likely to undertake different tasks). Without a structured or meaningful direction, agents might learn to differ in arbitrary or unproductive ways, which may not benefit multi-agent cooperation.
>
> We further compare the performance of our method against CIA [1], CTR [2], and TEE [3] in the SMAC-Exp benchmark, which is designed to evaluate the exploration capability of MARL algorithms in efficiently learning implicit multi-stage tasks, environmental factors, and micro-control. We examine our method in three hard tasks: Off_complicated, Off_hard, and Off_superhard. The experimental results are shown below. We note that our method maintains superior performance across all three tasks. Although these works are all based on contrastive trajectory representations, our proposed method using the inner-product-based intrinsic reward achieves more robust performance.
>
> | Methods | Off_complicated | Off_hard      | Off_superhard |
> |---------|-----------------|---------------|---------------|
> | CIA     | 0.13$\pm$0.07   | 0.47$\pm$0.11 | 0.03$\pm$0.02 |
> | CTR     | 0.42$\pm$0.19   | 0.32$\pm$0.06 | 0.12$\pm$0.07 |
> | TEE     | 0.64$\pm$0.16   | 0.56$\pm$0.09 | 0.06$\pm$0.04 |
> | DMD     | 0.78$\pm$0.13   | 0.72$\pm$0.05 | 0.37$\pm$0.18 |
>
>
>
> 3. Clarifications on the inner-product-based Wasserstein distance
>
> The inner-product intrinsic reward fundamentally enlarges the Wasserstein distance between the trajectory distributions of the current agent and other agents along diverse directions defined by $z_{j}$. Thus, the "significant variation" arises from combining two ideas:The agent must increase the magnitude of the Wasserstein distance, $||W(p_{\pi_{a}}, p_{\pi_{j}})||$. A larger Wasserstein distance by definition means the distributions $p_{\pi_{a}}$ and $p_{\pi_{j}}$ are "further apart" or more different. This creates diversity.The agent must change $p_{\pi_{a}}$ such that the direction of the difference, represented by the vector $W(p_{\pi_{a}}, p_{\pi_{j}})$, aligns with the fixed random direction $z_{j}$ for each agent. This creates structured variation.
>
> 4. New visualization results
>
> In our revised manuscript, we have added a UMAP visualization of trajectory representations in Figure 5. This visualization illustrates the alignment between these representations and the latent variables.

---

### Official Review · Reviewer_uRRB · 2025-11-01

**Soundness:** 3
**Presentation:** 3
**Contribution:** 3
**Rating:** 6
**Confidence:** 3

**Summary:**

The paper introduces an exploration method DMD based Wasserstein distance between different agents’ trajectory distributions in a latent trajectory representation space. Theoretical analysis and numerical examples are given to demonstrate the advantages of DMD.

**Strengths:**

The paper introduces an exploration method DMD based Wasserstein distance between different agents’ trajectory distributions in a latent trajectory representation space. Theoretical analysis and numerical examples are given to demonstrate the advantages of DMD over other alternatives in MARL.

**Weaknesses:**

The examples used are still small size. It is not clear how the proposed DMD method scale as the number of agents grow to larger size.

**Questions:**

What is the computational cost by DMD and whether it fits large scale muti-agent systems?

---

> ### Author Response · Authors · 2025-11-19
>
> Thank you for your careful review and for providing us with detailed and helpful feedback.
>
> To examine the scalability of our method, we evaluated it on a large-scale multi-agent benchmark, Magent. The Magent platform supports large-scale multi-agent reinforcement learning with tasks such as pursuit, battle, combined arms, and tiger deer. We tested our method on the battle task with varying numbers of agents. The evaluation results are shown in the table below. Compared to QMIX, our method continues to outperform it and scales well as the number of agents increases.
>
> | Number of agents |     QMIX     |      DMD     |
> |:----------------:|:------------:|:------------:|
> |        50        |  376$\pm$59 |  **273$\pm$73**  |
> |        75        | 892$\pm$228 |  **482$\pm$183**  |
> |        100       | 2759$\pm$695 | **1196$\pm$373** |
> |        125       | 3269$\pm$1427 |  **2375$\pm$562** |
>
> We next analyze the computational cost of our method. We note that compared to QMIX, our method needs to train the trajectory encoder and the dual functions to calculate the Wasserstein distance objective, which introduces additional computational overhead. We then present a comparison of training time between QMIX and our method in the three super hard scenarios of SMAC on the same computing platform in the table below. The results show that, compared to QMIX, our proposed method does not consume significant extra training time.
>
>
> |  Methods | Training time |            |                |
> |:--------:|:-------------:|:----------:|:--------------:|
> |          |   6h\_vs\_8z  |  corridor  | 3s5z\_vs\_3s6z |
> |   QMIX   |   8h 35m 29s  | 7h 17m 30s |   10h 42m 18s  |
> | DMD+QMIX |   8h 47m 36s  | 7h 26m 42s |   10h 50m 39s  |

---

### Note · Authors · 2025-11-29

I have read and agree with the venue's withdrawal policy on behalf of myself and my co-authors.